# Variations in Energetic Particle Fluxes around Significant Geomagnetic Storms Observed by the Low-Altitude DEMETER Spacecraft

Stefan Gohl [1,2,*], František Němec [2] and Michel Parrot [3]

1 Institute of Experimental and Applied Physics, Czech Technical University in Prague, 11000 Prague, Czech Republic
2 Faculty of Mathematics and Physics, Charles University, 18000 Prague, Czech Republic; frantisek.nemec@mff.cuni.cz
3 LPC2E/CNRS Orléans, 45071 Orléans, France; mparrot@cnrs-orleans.fr
* Correspondence: Stefan.Gohl@utef.cvut.cz

**Abstract:** A superposed epoch analysis is conducted for five geomagnetic storms in the years 2005 and 2006 with the aim to understand energetic particle flux variations as a function of L-shell, energy and time from the Dst minimum. Data measured by the low-altitude DEMETER spacecraft were used for this purpose. The storms were identified by a Dst index below $-100\,\mathrm{nT}$, as well as their being isolated events in a seven-day time window. It is shown that they can be categorized into two types. The first type shows significant variations in the energetic particle fluxes around the Dst minimum and increased fluxes at high energies ($>1.5\,\mathrm{MeV}$), while the second type only shows increased fluxes around the Dst minimum without the increased fluxes at high energies. The first type of storm is related to more drastic but shorter-lasting changes in the solar wind parameters than the second type. One storm does not fit either category, exhibiting features from both storm types. Additionally, we investigate whether the impenetrable barrier for ultra-relativistic electrons also holds in extreme geomagnetic conditions. For the highest analyzed energies, the obtained barrier L-shells do not go below 2.6, consistent with previous findings.

**Keywords:** geomagnetic storms; solar wind; earth's magnetosphere; radiation belt





## 1. Introduction

Disturbances in the Earth's magnetosphere are often caused by geomagnetic storms (e.g., [1]). These can have a serious impact on satellites as well as on power grids, communication and navigation on Earth. They are caused by solar wind-magnetosphere interaction through the magnetic reconnection mechanism [2–4]. This interaction causes enhanced energetic particle fluxes in the radiation belts [5–8] and an increased ring current encircling Earth [9,10]. The storms are characterized by a depression in the horizontal component of the geomagnetic field. The strength of the storms is often expressed in the disturbance storm time (Dst), index [11–13].

The sources of these solar-wind-driven interactions fall into two types. One type is interplanetary coronal mass ejections (ICMEs) (e.g., [14–17]), originating from coronal mass ejections (CMEs) at the Sun (e.g., [18]). The other type is so-called stream interaction regions (SIRs), also called corotating interaction regions (CIRs) (e.g., [19,20]), which cause recurring geomagnetic events. These are characterized by fast solar wind emanating from solar coronal holes interacting with the preceding ambient slower solar wind.

Research in recent years has been attempting to establish the ability to predict the geoeffectiveness of the impinging solar wind, i.e., to understand what solar wind parameters eventually result in significant geomagnetic storms and what the corresponding variations of energetic particles are in the Earth's magnetosphere. An investigation of a

large number of major geomagnetic storms (Dst $\leq -100\,$nT) from 1996 to 2005 and their solar and interplanetary sources revealed that the majority of storms are caused by one or multiple ICMEs, while only a small number of storms are caused by CIRs [21]. ICME-driven storms are brief, with denser plasma sheets, strong ring currents and more negative Dst. They pose more of a problem for Earth-based electrical systems. On the other hand, CIR-driven storms are of longer duration, with hotter plasmas, and they produce high fluxes of relativistic electrons [22]. They are more dangerous for space-based assets. However, not all ICMEs cause strong geomagnetic storms, as stated in [23]. ICMEs originating from active regions of the Sun cause strong storms with shorter transit times, while ICMEs originating outside of these regions are less geoeffective with longer transit times.

This study aims to contribute to this research by studying data from the Low-Earth-Orbit (LEO) DEMETER satellite. The goal is to understand the evolution of energetic particle fluxes in the Van Allen radiation belts at the times of significant geomagnetic storms. Furthermore, it is investigated whether all the geomagnetic storms behave in a similar manner or if some considerable differences between them can be identified. For this purpose, data from a LEO satellite has been used. The main advantage is the high sampling rate, because of their relatively short orbiting periods in the range of one or two hours. With satellites in a near-polar orbit, particle fluxes can be measured several times a day across all L-shells.

The used data set is described in Section 2. In Section 3, the results are presented, starting with the time evolution of the energetic particle fluxes as a function of L-shells and energy (Section 3.1). Variations in the solar wind parameters related to the storms are studied in Section 3.2. Finally, Section 3.3 investigates whether the impenetrable barrier for ultra-relativistic electrons, which effectively limits significant high energy particle fluxes to L-shells larger than about 2.8 [24,25], also holds in extreme geomagnetic conditions.

## 2. Data Set

The data set used in this study was provided by the IDP instrument [26] onboard the DEMETER spacecraft. DEMETER operated between June 2004 and December 2010 at a low-altitude orbit of initially 710 km. The altitude was changed to 660 km in December 2005. The spacecraft was on a quasi Sun-synchronous orbit with an inclination of 98.23°. The measurements took place either shortly before local noon (approx. 10:30) or shortly before local midnight (approx. 22:30). The IDP instrument consists of a fully depleted 1 mm thick silicon detector surrounded by an aluminium collimator with an opening angle of $\pm 16°$. A 6 μm thick aluminium foil stops parasitic light and protons below ~500 keV. The instrument was designed to primarily measure electron fluxes, but beyond the aluminium foil, there is no particle distinction. Therefore, we will only talk about particle fluxes instead of electron fluxes. The total deposited energy range from 70 keV to 2.34 MeV is divided into 255 channels plus one more channel to code for energy losses above 2.34 MeV.

There were two modes of operation, due to limited capacity of the telemetry. For the most part, the "survey mode" was active nearly continuously, collecting lower resolution data. In the "burst mode", high-precision data were taken at specific locations relevant for the mission objectives. This study uses data from the survey mode, as it provides a larger coverage in space and time. In the survey mode, the energy channels are grouped by two and the time resolution of the measurements is 4 s.

A comparison of the averaged particle fluxes for different energies over the L-shells was undertaken separately for the day and night side for the studied data set. It should be noted that data between the geomagnetic longitudes of $-30°$ and $120°$ are excluded in order to suppress the effects of the South Atlantic Anomaly. The longitudes were chosen according to Figure 2 in [27]. The results are shown in Figure 1, which depicts the average particles fluxes color coded as a function of energy (ordinate) and L-shell (abscissa). It can be seen that the fluxes below $L \simeq 6$ do not differ significantly between the day and the night side. The inner and outer radiation belts are clearly discernible, at least at lower particle energies. They are separated by an energy-dependent slot region located at

L-shells between about 2.5 and 3.5. At larger L-shells, deviations start to appear due to a considerably different magnetic field distortion at different local times, resulting eventually in drift-shell splitting [28] and more complicated L-shell definition [29]. We thus limit all the further analysis to $1 \leq L \leq 6$ and we do not distinguish between the day side and night side any longer.

Most significant geomagnetic storms are identified by a Dst index below $-100$ nT. Additionally, the events are required to be isolated in time, i.e., not interfering with other events in a seven-day window starting one day before the Dst minimum until six days after. This means that storms that showed a second Dst minimum during this time window were omitted from this study. This led to the selection of five storms. Four of the storms happened during the year 2005 between May and the end of August. The fifth storm was measured in December 2006. The exact days of the storms are summarized in Table 1.

**(a)** Day side                              **(b)** Night side

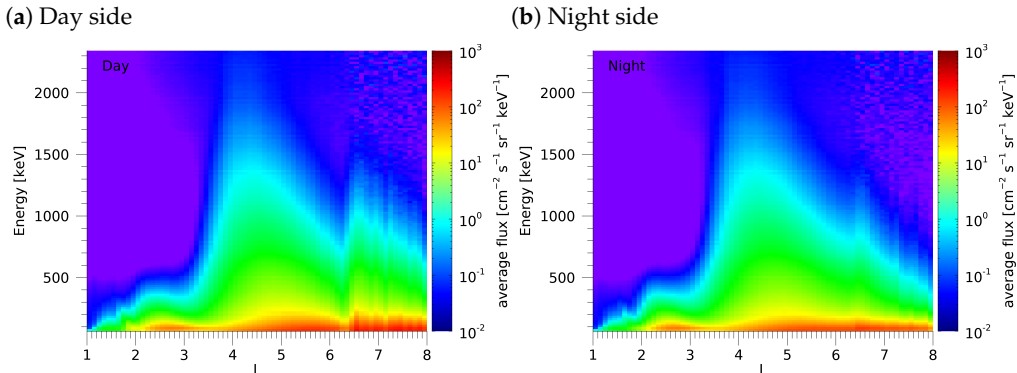

**Figure 1.** Average particle fluxes obtained during the entire duration of the DEMETER mission are color-coded as a function of energy (ordinate) and L-shell (abscissa) separately for the daytime (**a**) and nighttime (**b**).

**Table 1.** A list of five strong, isolated geomagnetic storms identified during the DEMETER mission analyzed in the present paper. Selection criteria are a Dst index below $-100$ nT and no other events in a seven-day period around the storm starting one day before the Dst minimum. The minimum Dst value and the type of the storm source are also displayed.

| # | Date | Dst Minimum [nT] | Type | Source |
|---|------|------------------|------|--------|
| 1 | 15 May 2005 | $-247$ | 1 | ICME |
| 2 | 13 June 2005 | $-106$ | 2 | ICME |
| 3 | 24 August 2005 | $-184$ | 1 | ICME/SIR |
| 4 | 31 August 2005 | $-122$ | 2 | SIR |
| 5 | 15 December 2006 | $-162$ | - | ICME |

## 3. Results

### 3.1. Energetic Particle Flux Evolution

Figure 2 shows a superposed epoch analysis of energetic particle flux variations during the five storms. The average fluxes are color-coded in a logarithmic scale as a function of L-shell (ordinate) and time (abscissa). The time of the Dst minimum is chosen as the reference point in time, and the time evolution is then plotted as a function of time relative to the Dst minimum. The fluxes are averaged over the entire energy range of the IDP instrument, i.e., between 70 keV and 2.34 MeV. Additionally, Figure 2 depicts the average Dst index over the course of the investigated time line. One can see that fluxes above $1\,(\mathrm{cm}^2\,\mathrm{s}\,\mathrm{sr}\,\mathrm{keV})^{-1}$ mostly occur after the onset of the main phase of the storms and above $L \simeq 2.2$. During the Dst minimum, occasional high fluxes are also observed at lower L-shells, but they dissipate already after about half a day after the Dst minimum. Fluxes above $L \simeq 2.2$ stay at a high level for a long time, and they only decrease slightly during the analyzed time interval.

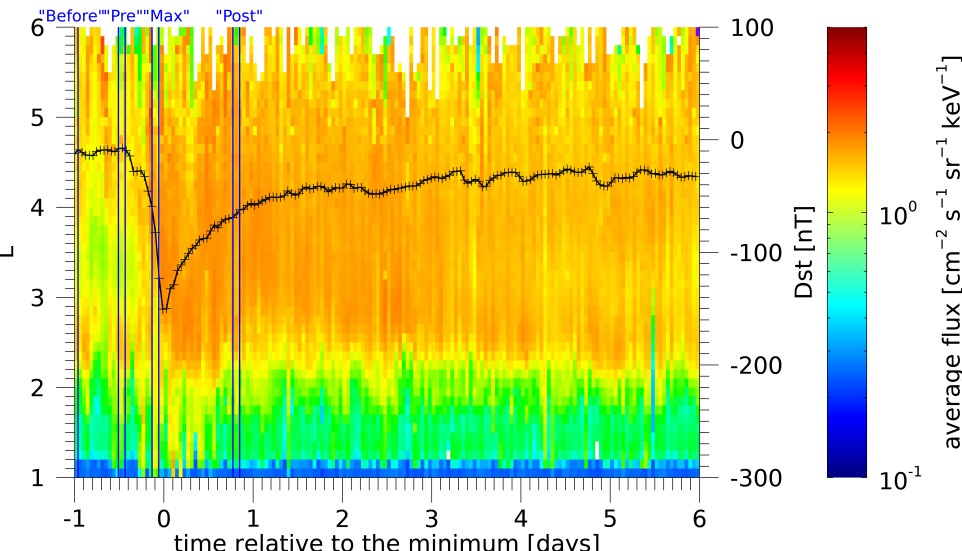

**Figure 2.** Superposed epoch analysis of all five geomagnetic storms in a seven-day window starting one day before the Dst minimum. The average energetic particle flux in the energy range between 70 keV and 2.34 MeV is color-coded as a function of the L-shell (ordinate) and time relative to the Dst minimum (abscissa). The average Dst index time dependence is overplotted by the thick black curve. There are four areas marked by vertical blue lines: one day "Before" the Dst minimum, at the Dst maximum preceding the storm onset ("Pre"), at the Dst minimum ("Max"), and one day after the Dst minimum ("Post").

There are four time subintervals marked by blue vertical lines in the plot, which are labeled "Before", "Pre", "Max", and "Post". Each of these time subintervals is two hours long. They are selected for a subsequent more detailed analysis. The time subinterval labeled "Before" starts one hour before the beginning of the plot and is therefore not fully visible.

Looking at the energetic particle flux dependences obtained for the individual storms, some systematic differences can be identified. The storms are categorized into two types according to their unique features presented in this section, labeled "Type 1" and "Type 2" storms, respectively. For each type, two storms could be identified. The fifth storm did not fit into either category, and it will be treated separately later in this section. Table 1 presents the five storms and their categorization. Furthermore, the source for each storm is displayed according to [30,31].

Type 1 storms are generally stronger and cause the Dst index to drop to lower values than Type 2 storms. Additionally, they are preceded by a well pronounced increase in the Dst index at the time of the interplanetary (IP) shock arrival before the storm onset itself. Type 2 storms show barely any change in the Dst index at this point.

Figure 3 presents the evolution of average fluxes and Dst index for both storm types using the same format as for Figure 2. The results obtained for Type 1 and Type 2 storms are depicted in Figure 3a,b, respectively. Type 1 storms show significantly increased fluxes at $L \geq 4$ well before the Dst minimum, essentially at the time when the interplanetary shock hits the magnetosphere. However, at the time of the Dst minimum, the fluxes at these high L-shells drop to about pre-storm levels. At the same time, fluxes rise to high levels at $2 \leq L \leq 4$. After about half a day, the fluxes acquire roughly the same levels throughout all $L \geq 2.2$. Type 2 storms show a somewhat different flux evolution. For these storms, no enhanced fluxes are observed before the storm onset. High fluxes at all L-shells above $L \simeq 2.2$ appear only around the Dst minimum.

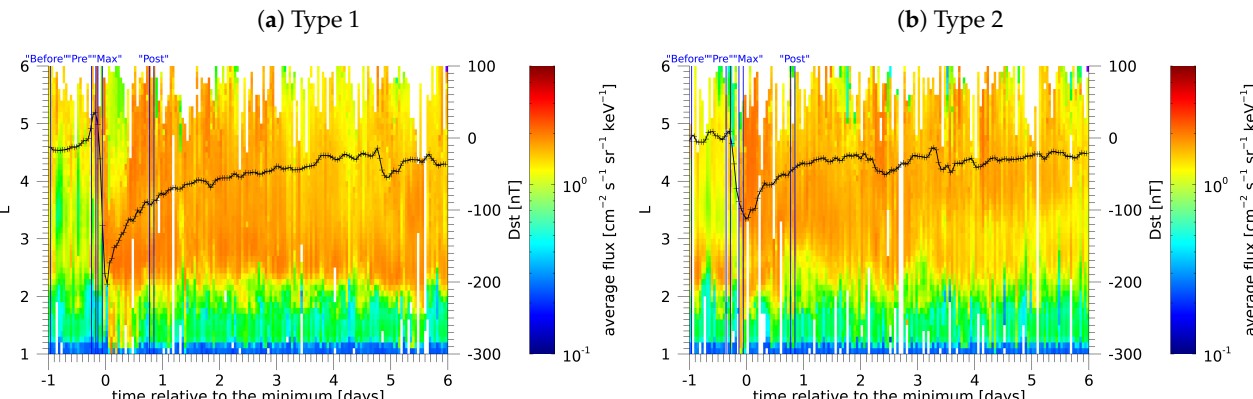

**Figure 3.** The same as Figure 2 but separated according to (**a**) Type 1 storms and (**b**) Type 2 storms.

Figure 4 shows the evolution of average fluxes in the L-shell range between 1 and 6 as a function of energy (ordinate) and time from the Dst minimum (abscissa). The results obtained for Type 1 and Type 2 storms are plotted in Figure 4a,b, respectively. The significant difference is the increased fluxes of energetic particles above about 1.5 MeV for Type 1 storms around the time of the Dst minimum. Note that, at the same time, the overall (energy-averaged) fluxes at L-shells larger than about 3.5 are severely depleted, as demonstrated by Figure 3a. Type 2 storms lack these enhanced energetic particle fluxes at this time.

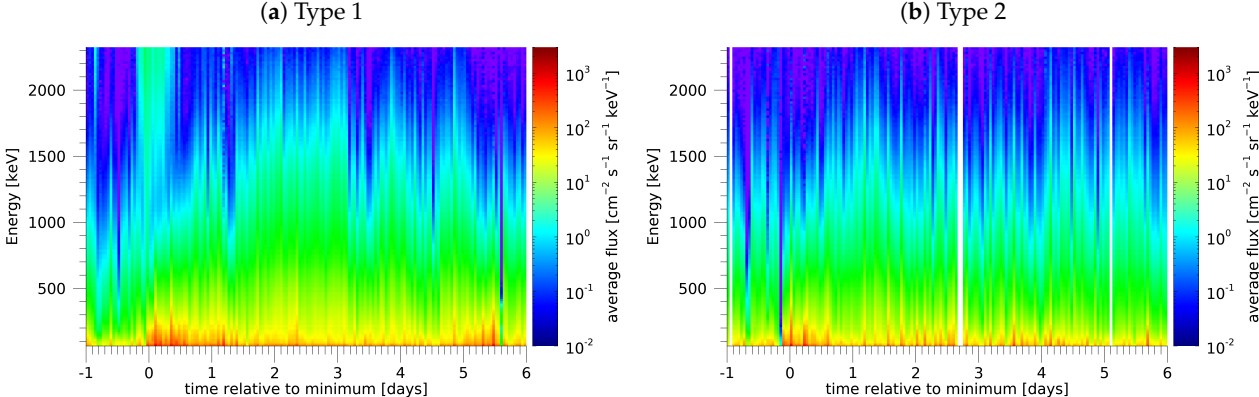

**Figure 4.** Average energetic particle fluxes in L-shell range between 1 and 6 as a function of energy (ordinate) and time relative to the Dst minimum (abscissa) for Type 1 storms (**a**) and Type 2 storms (**b**).

Figures 5 and 6 depict the average fluxes during the four previously selected time subintervals as a function of energy and L-shell for storm types 1 and 2, respectively. The plots show the average flux over both storms for each storm type. Each of the figure panels corresponds to one of the four time subintervals, starting with a period one day before the Dst minimum ("Before", Figures 5a and 6a), followed by the time interval around the maximum of the Dst index just before the drop ("Pre", Figures 5b and 6b). Figures 5c and 6c are around the Dst minimum ("Max"), and Figures 5d and 6d correspond to one day after the Dst minimum ("Post"). The most significant difference between the two storm types is visible during the storm maxima, where the fluxes are increased at all energies above $L \simeq 2.8$ for Type 1 storms. Some remnants are visible in the post-storm period with some high energy particles remaining at very high L-shells. While the energetic particle fluxes are considerably enhanced as well during Type 2 storms, the increase is less significant, in particular at the highest analyzed energies and larger L-shells.

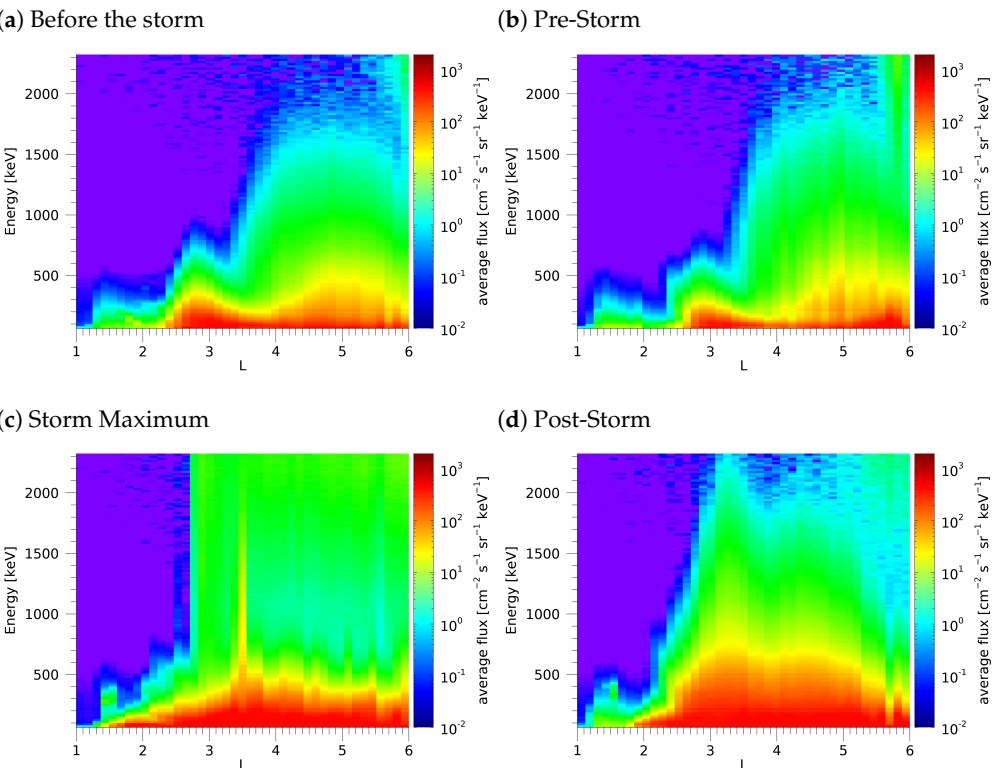

**Figure 5.** Average energetic particle fluxes during Type 1 storms are color-coded as a function of energy (ordinate) and L-shell (abscissa) during four selected two-hour-long time subintervals. (**a**) One day before the Dst minimum. (**b**) Around the Dst maximum just before the storm onset. (**c**) Around the Dst minimum. (**d**) One day after the Dst minimum.

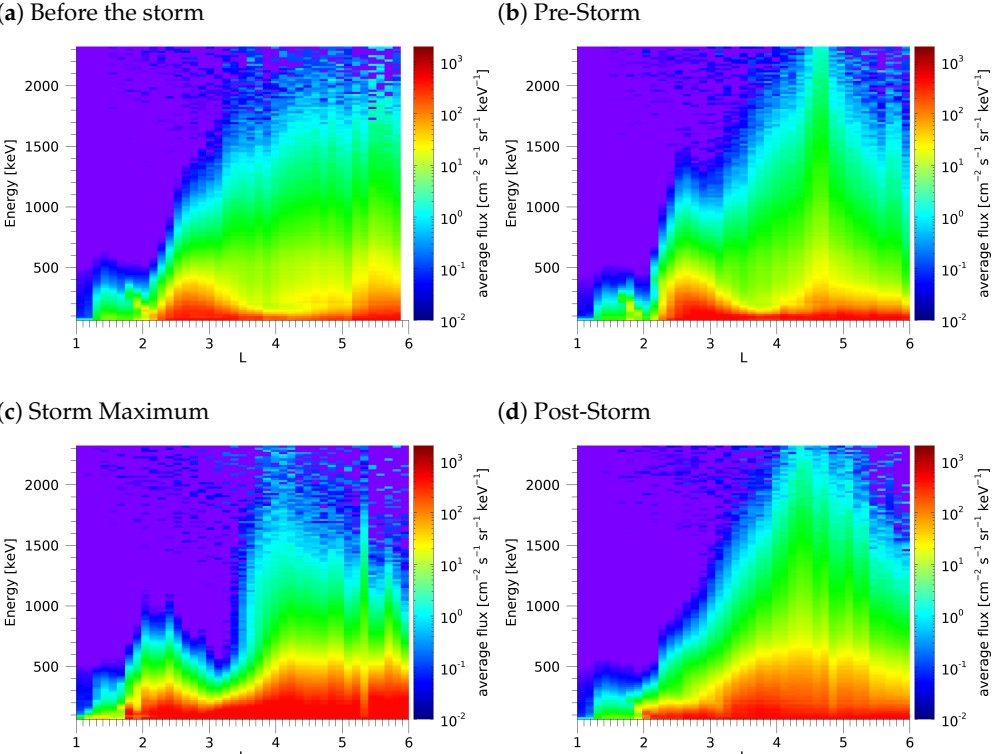

**Figure 6.** The same as Figure 5, but for Type 2 storms.

The results obtained for the fifth storm, which was categorized neither as Type 1 nor as Type 2, are shown in Figure 7 (called "special case" hereinafter). Figure 7a uses the same format as Figure 3 to depict the flux variations as a function of L-shell and the time relative to the Dst minimum. Figure 7b uses the same format as Figure 4 to depict the flux variations as a function of the particle energy and the time relative to the Dst minimum. It can be seen that, for this particular storm, the energetic particle fluxes are increased at L above 4 at high energies, similarly to the case of Type 1 storms. However, this already occurs about 16 h before the Dst minimum, i.e., earlier than for Type 1 storms. The drop in the fluxes at $L \geq 4$ around the Dst minimum is not visible, though. The behavior after the initial high fluxes more closely resembles the trend observed for Type 2 storms. The plots of the fluxes as a function of energy and L-shell obtained for the selected time subintervals are not shown for this event, as they are similar to Type 1 storm results. The only difference is that the energetic particle flux increases already in the pre-storm phase. The change in the Dst index lies in between Type 1 and Type 2 storms.

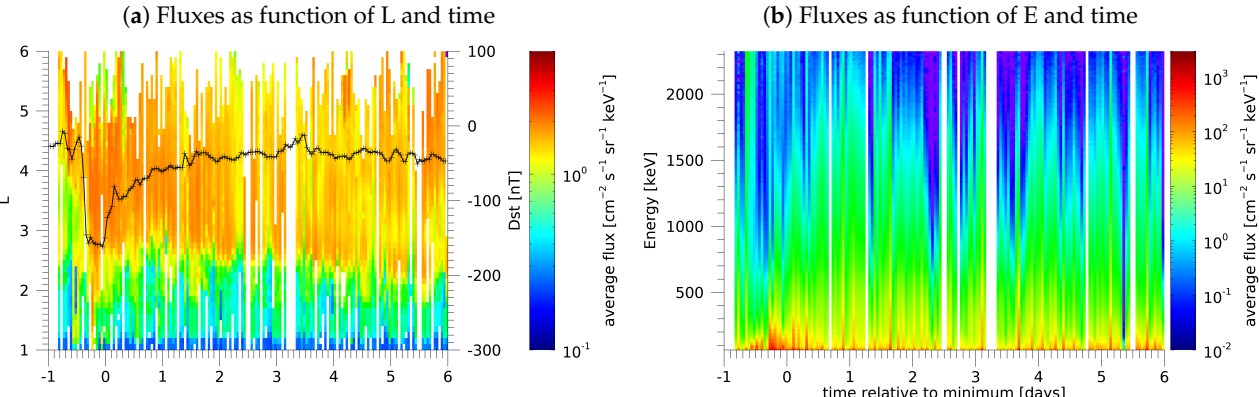

**Figure 7.** Dependences obtained for the fifth storm that fits neither Type 1 nor Type 2 classification. (**a**) Energetic particle fluxes color-coded as a function of L-shell (ordinate) and time from the Dst minimum (abscissa). The Dst index is overplotted by the thick black curve. (**b**) Average energetic particle fluxes at L-shells between 1 and 6 color-coded as a function of energy (ordinate) and time relative to the Dst minimum (abscissa).

### 3.2. Solar Wind Parameters

Now that energetic particle flux evolution during the individual storm types has been demonstrated, it is of interest to investigate how these are related to particular variations in the solar wind parameters. This is done in Figure 8, which depicts the time dependences of several selected solar wind parameters around individual storms, along with the Dst and AE index evolutions. Specifically, the solar wind flow speed $v$, plasma number density $n$, interplanetary magnetic field (IMF) magnitude $B$, IMF $B_z$ component and the plasma beta are plotted. The respective data for the solar wind parameters are taken from the OMNI data set [32] with 5 min time resolution. For each storm type, a representative storm has been chosen to showcase the typical solar wind parameter variations. Type 1 storms are generally stronger and the respective solar wind parameter changes are more sudden than for Type 2 storms. Both Type 1 storms happen to have a bipolar $B_z$, the south-to-north (SN) type in particular, while Type 2 storms and the special case are south (S) type storms.

In the case of Type 1 storms, the interplanetary shock arrives only shortly before the actual storm onset, as compared to Type 2 storms, where the related temporal/spatial scales seem to be much longer. The differences in the solar wind speed, the IMF magnitude $B$ and IMF $B_z$ reach significantly higher values for Type 1 storms than for Type 2 storms. Additionally, all these changes happen much faster for Type 1 than for Type 2 storms. For Type 1 storms, both the IMF and the solar wind plasma number density go back to normal values about half a day after the Dst minimum and the solar wind flow speed reaches pre-storm levels after two to three days. On the other hand, for Type 2 storms, the solar wind flow speed tends to keep rising until it reaches its maximum after a few

days. This was the case for both Type 2 storms. The IMF also takes longer to go back to its normal values, with IMF $B_z$ remaining negative for about a day after the Dst minimum. The plasma beta does not show any characteristics regarding Type 1 and Type 2 storms.

The solar wind parameters for the special case are mostly comparable with the solar wind parameters observed around the Type 2 storm times. The main difference seems to be the solar wind flow speed. Instead of a gradual rise observed for the Type 2 storms, it jumps suddenly up upon the IP shock crossing more than half day before the Dst minimum. Furthermore, it is noticeable that the AE index peaks prior to the drop in the Dst index, which is not visible for Type 1 nor Type 2 storms. This AE index peak appears to be concurrent with the IP shock arrival. It also coincides with the appearance of the high energetic particle fluxes at $L \geq 4$ observed about 16 h before the Dst minimum.

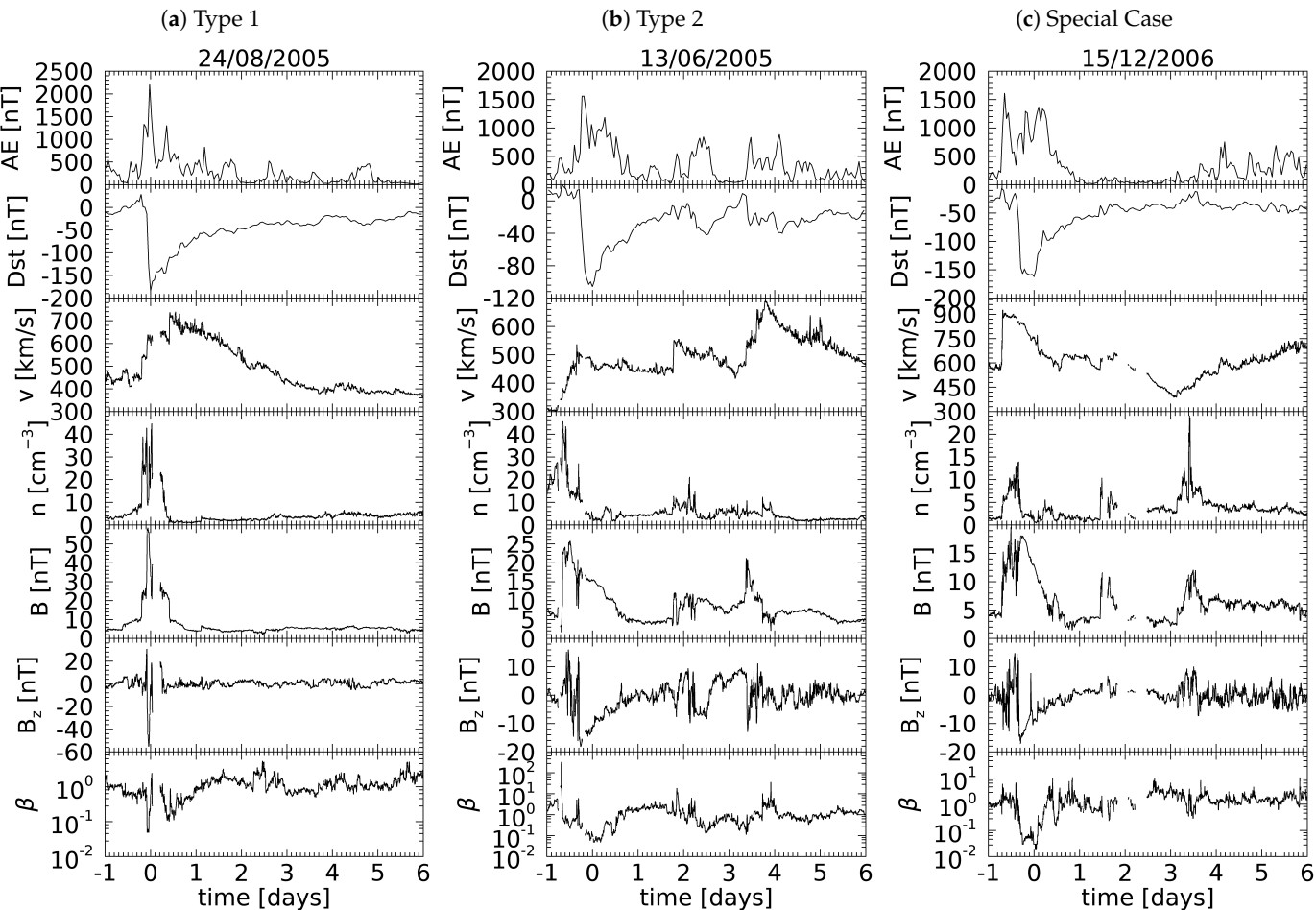

**Figure 8.** Time dependence of the AE and Dst indices, solar wind flow speed *v*, solar wind plasma number density *n*, IMF magnitude *B*, IMF $B_z$ component, and the plasma beta for representative storms of (**a**) Type 1 and (**b**) Type 2, as well as for (**c**) the special case.

### 3.3. L Barrier

An apparently impenetrable barrier for ultra-relativistic particles, effectively limiting significant high-energy particle fluxes to L-shells larger than about 2.8, was reported using the Van Allen Probes spacecraft data [24]. Consistent with these findings, our results also reveal a drastic drop in the flux from high to low L-shells. The issue is investigated more in detail in this section. In particular, the aim is to investigate the L-shell barrier location around the time of the storm maximum as a function of the energy, with the focus to check whether the L-barrier holds even during these extreme geomagnetic conditions. The analysis is done for the entire range of measured particle energies. However, in order

to suppress statistical fluctuations and to make the analysis more feasible, only eight energy bins are used to cover the entire energy range.

In order to determine the L-shell of the energetic particle flux drop, the average flux in a twelve-hour-long time interval starting at the Dst minimum is plotted as a function of L-shell. Generally, the fluxes start at nearly zero for low L-shells and rise significantly around a certain L, forming a steep flank. This flank is fitted with a hyperbolic tangent. The position of the L barrier is then defined as the point where the flank reaches half of its height. An example of the procedure applied to the highest energy bin (2057.6–2342.4 keV) and 15 December 2006 storm is shown in Figure 9. This is done for all five storms. The results obtained are presented in Figure 10. The different colors denote the different types of storms—blue for Type 1 storms, green for Type 2 storms, and red for the special case. For energies around 100 keV, all storms show a barrier between $2.2 \leq L \leq 2.4$. However, the respective barrier locations split significantly towards higher energies. Above 1 MeV the barrier locations stay more or less constant at L-values between about 2.5 and 3.7. It is noteworthy that for the highest energies analyzed, the obtained barrier L-shells do not get below about 2.6, making them consistent with former results [24]. Additionally, it would appear that the barrier L-shells during Type 2 storms are generally higher than during the other storm types. This may be interpreted as consistent with Type 2 storms being more gradual and generally weaker.

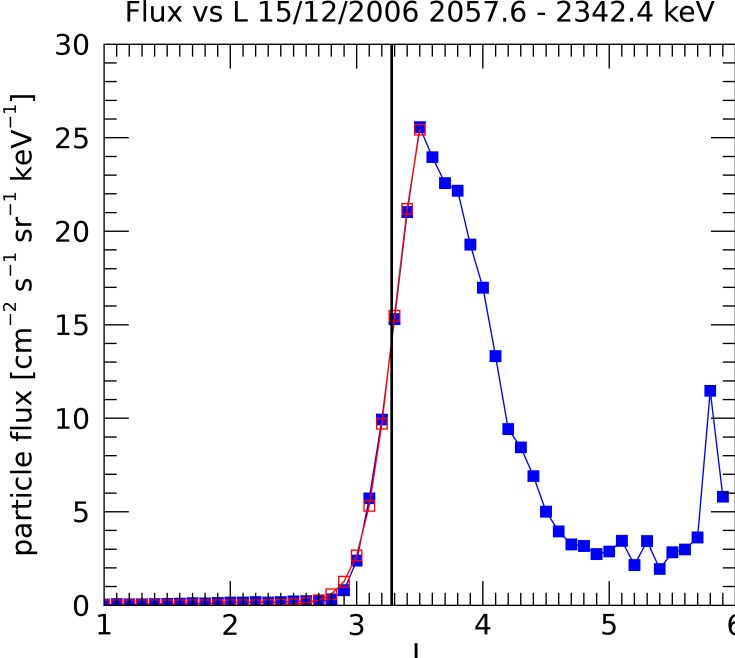

**Figure 9.** The particle flux measured during the 15 December 2006 storm in the energy range 2057.6–2342.4 keV (blue). The left-most flank is fitted with a hyperbolic tangent (red). The position of the L barrier is defined as the point, where the flank reaches half of its height. This position is marked by the vertical black line.

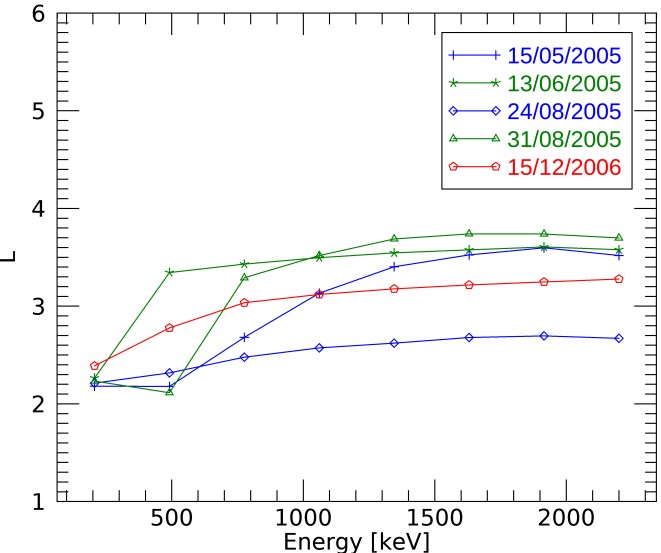

**Figure 10.** The impenetrable barrier L-shells as a function of energy for all five storms. The colors represent the different storm types, blue being Type 1 storms, green Type 2 storms, and red corresponds to the special case.

## 4. Discussion and Conclusions

Energetic particle data measured by the IDP instrument onboard the DEMETER spacecraft have been analyzed in terms of geomagnetic storms between June 2004 and December 2010. The selection criteria were a Dst index of $-100$ nT or below and no other apparent event in a seven-day-long time window starting one day before the Dst minimum. This led to five storms being investigated, most of them from 2005 and one from 2006. It was shown that the storms can be classified into two categories based on different energetic particle flux variations around the storm maximum, denoted Type 1 and Type 2. For each type, two storms could be identified. The fifth storm did not fit in any of these types.

Type 1 storms are characterized by the appearance of high fluxes just before the drop in the Dst index at $L \geq 4$. These fluxes decrease significantly shortly after the onset of the storm, while at $2.2 \leq L \leq 4$, the fluxes increase significantly. At the same time, increased fluxes of energetic particles ($> 1.5$ MeV) appear across all $L \geq 2.2$. Type 2 storms do not feature energetic particles throughout the observed period in this fashion. The fluxes increase only once the storm approaches the Dst minimum, and they increase across all $L \geq 2.2$. The Dst index reaches lower values for Type 1 storms than for Type 2 storms.

Another difference between the individual storm types was revealed by the investigation of the solar wind parameters during the respective periods. The onset of the Type 1 storms typically follows shortly after the arrival of an IP shock. This is accompanied by significant sudden and rather short-lasting variations in the solar wind parameters, in particular the IMF $B_z$ component, which turned highly negative for a limited period of time. On the other hand, the time delay between the Type 2 storms onset and the IP shock arrival is comparatively longer. Additionally, although the IMF $B_z$ values eventually turn negative as well, they never become as low as for Type 1 storms. Nevertheless, they stay negative for a longer time period, for about a day.

The special case has increased fluxes at $L \geq 4$ already 16 h before the Dst minimum. This resembles the behavior characteristic pf the Type 1 storms, just at earlier times. Otherwise, the fluxes around the Dst minimum and the solar wind parameters show similar behavior as observed for the Type 2 storms. There is a peak in the AE index about 16 h before the Dst minimum, coincident with the IP shock arrival and with the appearance of the high fluxes of energetic particles at high L-shells. This peak is not present in any of the other storms.

Even though only five strong isolated geomagnetic storms suitable for the analysis were identified over the entire duration of the DEMETER mission, two distinctive categories were found. One type features significant variations in the energetic particle fluxes around the Dst minimum with increased fluxes at high energies (>1.5 MeV), while the other type shows the increased fluxes only at comparatively lower energies. Type 1 storms in this study only seem to be SN-type storms with higher geoeffectiveness, while Type 2 storms are the S-type storms with lower geoeffectiveness. This seems to contradict previous findings [33]. However, it might be related to the strength of the storms, as the IMF $B_z$ component for Type 1 storms is much more negative than for Type 2 storms. To verify this, more storms must be investigated. Overall, the behavior of energetic particle fluxes at the times of geomagnetic storms is highly complicated [1], and a conclusive reason for the distinctive behavior of Type 1 and Type 2 storms, as well as for the special case, was not found so far.

Finally, an investigation of the impenetrable barrier for particles was undertaken across all measured energies for the five individual storms during the period of the strongest disturbance. We showed that the apparent barrier is identifiable over a wide range of energies. For a given storm, the barrier location tends to shift toward larger L-shells for more energetic particles, but it ultimately stays nearly constant for energies above about 1 MeV, consistent with former findings [24].

**Author Contributions:** Conceptualization, S.G. and F.N.; methodology, S.G. and F.N.; validation, F.N. and M.P.; formal analysis, S.G.; investigation, F.N. and M.P.; resources, F.N. and M.P.; data curation, S.G. and F.N.; writing—original draft preparation, S.G.; writing—review and editing, S.G., F.N. and M.P.; visualization, S.G.; supervision, F.N.; project administration, F.N.; funding acquisition, F.N. All authors have read and agreed to the published version of the manuscript.

**Funding:** This research was funded by GACR Grant 21-01813S.

**Institutional Review Board Statement:** Not applicable.

**Informed Consent Statement:** Not applicable.

**Data Availability Statement:** The DEMETER data is available at https://sipad-cdpp.cnes.fr (last accessed on 21 July 2021). The OMNI data is available at https://omniweb.gsfc.nasa.gov (last accessed on 21 July 2021).

**Acknowledgments:** We thank the engineers from CNES and scientific laboratories (CBK, IRAP, LPC2E, LPP, and SSD of ESTEC) who largely contributed to the success of the DEMETER mission. DEMETER data are accessible from the https://sipad-cdpp.cnes.fr website (last accessed on 21 July 2021). Furthermore, the OMNI data were obtained from the GSFC/SPDF OMNIWeb interface at https://omniweb.gsfc.nasa.gov (last accessed on 21 July 2021).

**Conflicts of Interest:** The authors declare no conflict of interest.

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
