# Peer review of "Variations in Energetic Particle Fluxes around Significant Geomagnetic Storms Observed by the Low-Altitude DEMETER Spacecraft"

_universe, doi:10.3390/universe7080260_

Round 1

Reviewer 1 Report

see attached file

Reviewer 2 Report

Comments to the manuscript universe-1284039-peer-review-v1

“Variations of energetic particle fluxes around significant geomagnetic storms observed by the low-altitude DEMETER spacecraft”

by Stefan Gohl, František Němec and Michel Parrot

Since there are no two absolutely similar disturbances, the search for any common properties, the classification of types are of particular importance. In the presented manuscript, this refers to the behavior of the Dst index and the relationship with it of variations in fluxes of energetic particles measured on the DEMETER spacecraft. The authors managed to distinguish two types of storms with two cases of similar variations and one special type. Differences in variation for different types were found in all two-hour periods tied to the time of the Dst minimum as the reference point in time. An important fact is the discovery of the distinctive features of the behavior of the solar wind parameters and the geomagnetic indices AE and Dst for each type of storms.

The results of the paper are an example of the fact that the data of past missions and years cannot be outdated, having great scientific potential.

The work is of interest to the readers of the journal, is easy to read and can be recommended for publication with practically no changes.

I would like to point out a few unimportant points.

1.Although the authors note that a conclusive reason for the distinctive behavior of Type 1 and Type 2 storms was not found so far, maybe they can say something why the impenetrable barrier for type 1 cases is so different?

  1. Line 122: You can add a decryption of the abbreviation IP.
  2. Line 229: being well consistent with former results - with whose results?

Reviewer 3 Report

The article analyzes particle fluxes variations for 5 different storms measured by the DEMETER satellite. The paper is interesting, but it needs some corrections/clarifications.

1) Paragraph 2:

The authors cite a paper for the description of IDP. Nevertheless, for the readability of the paper, I think it is important to mention here a few basic features of the instrument: particle sensibility (electrons, protons, nuclei…), discrimination capabilities, relative energy window per particle type, energy reconstruction, angular aperture.. Also the survey mode could be better described.

2) Sentences 82-82: the authors could better clarify the reason of the differences in Figure 1 between day and night side, regarding L-shell >6.

3) Figure 1 should be moved to page 2, Figure 2 to page 3. As it is now, the readability of page 4 is really difficult.

4) Figure 3: a similar picture for quiet time periods would help seeing the flux increase. The “Before” status in the figure is already perturbed, so better if we could compare really quiet time periods with Figure 3.

5) Figures 3, 4, 7 are totally out of the margins, including captions. They should be better formatted.

6) Paragraph 3.3

The first sentence seems to be cut, does not make sense as it is written now.

7) In general, in this paragraph 3.3 the authors suddenly speak about “electrons”, while so far all pictures and sentences seemed to be composed by “all particle spectra”. Do you select electrons for this L barrier analysis? And how then?

8) Line 220: the selection of the L barrier is not shown, why? At least for one specific storm, it would be useful to follow the steps of the analysis.

9) Paragraph 4

Line 234; substitute “has” with “have”.

10) Lines 242-249: the authors again speak about electrons in this section, while the same sentences in the section “Results” indicated always “particles”. It is quite confusing, the reader does not proper understand if this analysis is integrated over all particles or is only for electrons.

Please clarify this issue.

Round 2

Reviewer 1 Report

See attachment
